# *C*_2_-Symmetric P-Stereogenic Ferrocene Ligands with Heavier Chalcogenophosphinous Acid Ester Donor Sites [note 1]

**DOI:** 10.3390/molecules26071899

**Published:** 2021-03-27

**Authors:** Roman Franz, Clemens Bruhn, Rudolf Pietschnig

**Affiliations:** Institute for Chemistry and CINSaT, University of Kassel, Heinrich Plett-Straße 40, 34132 Kassel, Germany; roman.franz@uni-kassel.de (R.F.); bruhn@uni-kassel.de (C.B.)

**Keywords:** ferrocene, phosphorus, chalcogene, bond activation, multinuclear NMR

## Abstract

*tert*-Butyl-substituted diphospha[2]ferrocenophane was used as a stereochemically confined diphosphane to investigate the addition of various dichalcoganes (R2Ch2; Ch = S, Se, Te and R = Me, Ph). Bischalcogenophosphinous acid esters bearing four soft donor sides were obtained as a mixture of *rac* and *meso* diastereomers and characterized by means of multinuclear NMR and X-ray analysis. The coordination chemistry of multidentate ligand **3b** was explored toward d^10^ coinage metal centers (Cu(I), Ag(I), and Au(I)), yielding various bimetallic complexes.

## 1. Introduction

Phosphinite ligands are well established in coordination chemistry and catalysis, featuring useful properties between phosphanes and phosphites in terms of donor–acceptor characteristics [1]. Similarly, thio-, seleno-, and tellurophosphinous acid esters have been reported, which can be formally regarded as isomers of the related thio-, seleno-, and tellurophosphoranes [2,3].

In general, chalcogeno phosphinous acid esters are prepared from diorganohalophosphanes by reaction with the corresponding metalchalcogenolate [1,4]. As an alternative, oxidative cleavage of a diphosphane with dichalcoganes (alias dichalcogenides) has also been reported [2,3]. The latter approach was employed recently by Hey-Hawkins and coworkers in preparing a set of carbaborane-bridged bischalcogenophosphinous acid esters with sulfur or selenium [5]. In the context of our ongoing investigation of > P-P < bond activation of phosphorus-rich ferrocenophanes [6,7], we wondered whether a similar ferrocene-bridged system may be accessible starting from [2]ferrocenophane **1**. Given the relevance of ferrocene-based ligand systems [8,9,10], we set out to explore the preparation of 1,1′-ferrocenylene-bridged bischalcogenophosphinous acid esters with sulfur, selenium, and tellurium. Owing to the vicinity of two adjacent soft donor sites, phosphorus and chalcogen, we explored the coordination behavior of these ligands toward d^10^ coinage metal centers.

## 2. Results and Discussion

### 2.1. Ligand Preparation and Properties

To study the addition of dichalcoganes to stereochemically predefined diphosphane, we chose [2]ferrocenophane **1**, which occurs exclusively in the *rac* form as the *meso* form is disfavored owing to repulsive steric interaction of the *tert*-butyl substituents at phosphorus [6]. Reaction of **1** with diphenyldisulfide, -selenide, and -telluride afforded the respective bischalcogenophosphinous acid esters **3a–c** (**a** = S, **b** = Se, and **c** = Te) via formal addition to the central P-P bond (Scheme 1). Despite the stereochemically preorganized relative configuration of the P-stereogenic centers in **1**, the stereoinformation got lost during the reaction, and **3a**–**c** were obtained as mixtures of their *rac* and *meso* diastereomers. The same products were obtained when starting from lithium bisphosphanide **2** (and diphenyldisulfide or diphenyldiselenide), in which the phosphorus atoms adopted a prochiral situation owing to doubly bridged metal coordination, as previously established [11,12]. In the latter reaction, two equivalents of the dichalcogenide were required, making it less efficient in terms of chalcogene transfer as one equivalent was eliminated as lithium chalcogenolate (Scheme 1). The loss of stereoinformation might be attributed to a stepwise transfer of the chalcogenyl units. Moreover, variable temperature (VT)-NMR investigations on closely related ferrocenophanes revealed low inversion barriers at phosphanyl units adjacent to ferrocene [7,13]. The latter explanation would be consistent with findings for carbaborane-bridged analogs, where a preference for the *rac* diastereomer was observed [5], which might be a consequence of the different electronic properties of the bridging unit affecting the inversion barrier at phosphorus. The stepwise transfer of chalcogenyl units could also explain the formation of [2]ferrocenophane **1** after the elimination of lithium phenyl tellurolate when **2** was treated with diphenylditelluride.

Apart from the chalcogeno phosphinous acid phenylesters described above, we were also interested in exploring the corresponding methyl esters as the smallest alkyl derivatives. To this end, we reacted [2]ferrocenophane **1** with dimethyl disulfide and dimethyldiselenide. Again, the corresponding thio- and selenophosphinous acid esters **4a**–**b** (**a** = S and **b** = Se) were obtained in ca. 80% yield as mixtures of *rac* and *meso* diastereomers (Scheme 2). For compounds **3a**–**c** and **4a**–**b**, the molecular structure in solid state was determined using single-crystal X-ray crystallography (Figure 1 and Figure 2). While chalcogenophosphinous acid esters have been known for several decades, to our knowledge, no solid-state structure of a telluro phosphinous acid ester has yet been reported in the literature.

Thio phosphinous acid ester **3a** crystallized with two molecules in the asymmetric unit in the monoclinic space group *P*2_1_/*c*, while heavier phosphinous acid esters **3b** and **3c** crystallized in the triclinic space group $P\overset{-}{1}$. While **3a** showed staggered (torsion angle *τ* between Cp-Cp(centroid)-Cp(centroid)-Cp was about 158°) conformation of the Cp units, the Cp rings of **3b** (*τ* = 72°) and **3c** (*τ* = 70°) were oriented in the eclipsed form. The sum of C-P-C and C-P-Ch angles around P1 slightly increased in the series **3a**–**c** from 301.28(8)° in **3a** to 305.80(4)° in **3c**. The Ch–P distances in **3a**–**c** (**3a**: S1-P1 2.1236(8) Å, **3b**: Se1-P1 2.2595(11) Å, and **3c**: Te1-P1 2.4660(2) Å) were typical for chalcogenophosphinous acid esters containing three-coordinate phosphorus [5,14,15,16,17]. Bisthiophosphinous acid methylester **4a** and bisselenophosphinous acid methylester **4b** crystallized in the orthorhombic space group *Pbcn* with a staggered conformation (*τ* = 90° in **4a** and *τ* = 88° in **4b**) of the ferrocene moiety. The Ch–P distances (**4a**: S1-P1 2.1119(13) Å and **4b**: Se1-P1 2.2581(17) Å) in **4a** and **4b** and the sum of C-P-C and C-P-Ch angles around P1 (**4a**: 302.31(13)° and **4b**: 303.30(20)°) were almost identical to those observed in **3a** and **3b**.

The chalcogeno phosphinous acid esters described above featured characteristic multinuclear NMR spectra. Upon ring cleavage of the [2]ferrocenophane scaffold, the ^31^P NMR chemical shift of 20.6 ppm in **1** [18] changed to lower field for the chalcogeno phosphinous acid esters **3a**–**c** and **4a**–**b** (Table 1). Two sets of signals (Δδ(^31^P) = 0.3–1.0 ppm) could be observed in the NMR spectra due to the fact that both diastereomers (*rac* and *meso*) were formed in nearly equal amounts starting from sterically confined **1** or prochiral **2**.

An interesting feature was observed for chalcogeno phosphinous acid esters **3b**, **3c**, and **4b**, which showed signal splitting of the selenium or tellurium satellites in the case of one diastereomer (Figure 3). This signal splitting might indicate intramolecular Ch···Ch interactions in solution, which, for steric reasons, should only be present in the *meso* form. The latter observation is also consistent with ^77^Se NMR spectra of **3b** and **4b** showing two sets of signals (**3b**: 216.2 (d) and 216.4 ppm (dd) and **4b**: −21.5 (d) and −21.8 ppm (dd)) for the *rac* and *meso* diastereomers, which split into a doublet (**3b**: ^1^*J*_SeP_ = 225 Hz and **4b**: ^1^*J*_SeP_ = 218 Hz) and a doublet of doublets (**3b**: ^1^*J*_SeP_ = 225 Hz, ^n^*J*_SeP_ = 5 Hz and **4b**: ^1^*J*_SeP_ = 218 Hz, ^n^*J*_SeP_ = 6 Hz), indicating ^1^*J*_SeP_ and ^2^*J*_SeP_ coupling. An analogous situation was found in ^125^Te NMR of **3c**, where two resonances were found at 288.2 (d, ^1^*J*_TeP_ = 497 Hz) and 284.2 (dd, ^1^*J*_TeP_ = 497 Hz, ^1^*J*_TeP_ = 5 Hz) ppm, which showed coupling to just one or both phosphorus nuclei depending on the nature of the isomer. Moreover, an intermolecular Ch···Ch interaction in **3b** could be excluded by a diffusion ordered spectroscopy NMR (DOSY-NMR) experiment (Appendix A), which showed no aggregate formation in solution. In the literature, homo- and heteronuclear interactions involving heavier chalcoganes have both been described even for the gas phase [19]. We cannot unambiguously conclude whether a direct P···Ch or a Ch···Ch interaction was operative in the case of esters **3b**, **3c**, and **4b**, but based on the unpeculiar ^31^P NMR chemical shifts of the *meso* forms of these compounds, a Ch···Ch interaction definitely seems more likely.

Besides NMR spectroscopic characterization, the identity and purity of all newly prepared compounds were further established with mass spectrometry and elemental analysis.

### 2.2. Coinage Metal Complexes

To explore the ligand properties of the seleno derivative **3b**, its coordination behavior toward d^10^ coinage metal centers (Cu(I), Ag(I), and Au(I)) was explored, which should allow characterization of the resulting products with NMR spectroscopy. Despite the presence of two adjacent soft donor centers (P and Se) in the investigated ligand, coordination at the phosphorus lone pair was preferred exclusively in all cases (Scheme 3). In the case of copper(I) and silver (I), **3b** acted as a chelating ligand, forming metalla-bridged [3]ferrocenophanes **5** and **6**, where one acetonitrile completed the coordination sphere of the d^10^ metal. Forming such a structural motif, in principle, two diastereomers might be anticipated, where the lone pairs of the phosphorus atoms point to the same side (*meso* form) or opposite sides (*racemic* form) of the ring. In the case of **5** and **6** exclusively, the *racemic* form was observed, indicating epimerization of a stereocenter in the *meso* ligand during complexation, which proceeded quantitatively in the case of **6**. When ligand **3b** was reacted with two equivalents of gold(I) chloride, a dinuclear gold complex is formed as a mixture of *rac* and *meso* diastereomers. In the latter motif, the repulsive steric interaction of the substituents at phosphorus could be avoided more easily than in the mononuclear complexes **5** and **6**.

Upon coordination to copper(I), the ^31^P NMR chemical shift of the set of signals at 57.1 and 57.8 ppm in **3b** changed to higher field. Chemical equivalence of the ^31^P nuclei in **5** resulted in only one broad signal at 51.6 ppm. Contrary to that, the coordination of phosphinous acid ester **3b** to silver(I) changed the ^31^P NMR chemical shift to lower field (75.6 ppm) in **6** and signal splitting owing to coupling to ^107^Ag (^1^*J*_AgP_ = 401 Hz) and ^109^Ag (^1^*J*_AgP_ = 462 Hz) nuclei was observed. The ^77^Se NMR spectra of the copper(I) and silver(I) complexes showed higher-order spectra with a chemical shift at 230.5 ppm for **5** and 227.1 ppm for **6** resonating at lower field compared to the free ligand. To determine the phosphorus–selenium coupling constants, the ^77^Se NMR spectra were simulated (Figure 4and Appendix A). The values of the coupling constants to the directly bonded phosphorus nuclei (^1^*J*_SeP_ = 276 Hz in **5** and ^1^*J*_SeP_ = 286 Hz in **6**) were similar and slightly higher than in **3b** (^1^*J*_SeP_ = 225 Hz).

The coordination of **3b** to gold(I) changed the ^31^P NMR shift to lower field and resulted in two singlet signals at 85.9 and 85.2 ppm in **7** with selenium satellites (^1^*J*_SeP_ = 329 Hz and ^1^*J*_SeP_ = 332 Hz, respectively). ^77^Se NMR spectra of compound **7** showed a set of two doublets at 271.1 (^1^*J*_SeP_ = 329 Hz) and 268.9 (^1^*J*_SeP_ = 332 Hz) ppm for the Se atoms of the two different *rac* and *meso* diastereomers, which were deshielded compared to **5** and **6**.

Copper(I) complex **5** and silver(I) complex **6** crystallized with two molecules in the asymmetric unit in the triclinic space group $P\overset{-}{1}$, while the gold(I) complex **7** crystallized in the monoclinic space group *P*2_1_/c. The X-ray crystal structures of **5** and **6** (Figure 5) revealed a trigonal planar coordination geometry (**5**: ΣCu1 = 358.9(1)° and **6**: ΣAg1 = 357.6(1)°) at the coinage metal atom with d^10^-metal atoms bonded to two phosphorus units with bite angles between 113.82(2)° in **6** and 115.36(5)° in **5**, which are comparable to those found in related dppf analogs [20,21,22]. The tilt angle (α) of the ferrocene unit showed values of 2.3(2)° (**5**) and 2.25(14)° (**6**), which is the typical range for phosphorus-rich [3]ferrocenophanes [6,7,11,12,13,23]. The torsion angle *τ* increased from 49° in **5** to 55° in **6**, which indicated a high distortion of the ferrocene unit. The preferred linear coordination of gold(I) prevented the formation of a gold(I)-bridged [3]ferrocenophane. The resulting dinuclear complex **7** (Figure 6) entailed two Au-Cl units that were almost linear (P1-Au1-Cl1 179.12(4)° and P2-Au2-Cl2 176.83(3)°) with P-Au distances of 2.2277(90) Å (P1-Au1) to 2.2307(9) Å (P2-Au2). The P-Se bond lengths remained unchanged after coordination to the transition metal centers (**5**: 2.2556(14) Å, **6**: 2.2523(7) Å, and **7**: 2.2541(10) Å) compared to the free ligand **3b** (P1-Se1 2.2595(10) Å).

To investigate the possibility of achieving the same coordination motif as for copper(I) and silver(I) bisselenophosphinous acid phenylester, **3b** was reacted with one equivalent (tht)AuCl, which resulted in the formation of various undefined products besides unreacted free ligand. If the same reaction was repeated in the presence of aluminum chloride as Lewis acid, dimeric ionic gold(I) complex **8** (Scheme 4) was formed in good yield. Dimerization entails a linear coordination and a stabilization of both gold(I) centers in the ferrocenophane.

The dinuclear gold(I) complex **8** described above featured characteristic multinuclear NMR spectra. Upon coordination to gold(I), the ^31^P NMR chemical shift of the set of signals at 57.1 and 57.8 ppm in **3b** changed to lower field. Chemical equivalence of the ^31^P nuclei in **8** resulted in only one signal at 91.0 ppm, which was almost unchanged compared to the gold(I) complex **7**.

The corresponding ^77^Se NMR resonance of **8** at 275.3 ppm was deshielded by more than 50 ppm upon coordination to gold(I) compared to **3b**, while the ^1^*J*_SeP_ coupling dropped to 177 Hz. The resulting ^77^Se NMR spectrum was again a higher-order spectrum due to the presence of four selenium nuclei in compound **8**. Gold(I) complex **8** crystallized in the triclinic space group $P\overset{-}{1}$. The X-ray crystal structure of **8** (Figure 6) revealed that the gold atoms were bonded to two phosphorus units. The tilt angles α showed values of 3.9(1)° at Fe1 and 5.8(1)° at Fe2 for both ferrocene units and indicated that almost no ring strain was present. The Au-P contacts between 2.305(10) Å and 2.337(9) Å were slightly longer than those observed in **7** but in the typical range for P-Au bond lengths [24,25]. For the solid-state structures of compounds **7** and **8**, no inter- or intramolecular Au···Au interactions were observed (Au···Au > 6.7 Å in **8**). Nevertheless, for related dinuclear Au(I) complexes of ferrocene-bridged bisphosphanes, Au···Au distances were found to be quite variable in different polymorphs [26].

## 3. Materials and Methods

### 3.1. Experimental

All reactions were carried out by means of standard Schlenk or glovebox techniques under inert gas atmosphere (argon). Solvents were dried over Na/K alloy before use and were freshly distilled under inert gas. Deuterated solvents for NMR spectroscopy were dried and stored over molecular sieves. [Fe(C_5_H_4_P(*t*Bu)Li)_2_] and [Fe(C_5_H_4_P(*t*Bu)_2_] were prepared according to procedures in the literature [6,11], while other reagents were used as received without further purification. ^1^H-, ^13^C-, ^31^P-, ^77^Se-, ^125^Te-, and DOSY-NMR data were recorded on Varian VNMRS-500 MHz or MR-400 MHz spectrometers at 25 °C. Chemical shifts were referenced to residual protic impurities in the solvent (^1^H) or the deuterio solvent (^13^C) and reported relative to external SiMe_4_ (^1^H, ^13^C), H_3_PO_4_ (^31^P), Me_2_Se (^77^Se), or Me_2_Te (^125^Te). For molecular weight estimation, external calibration curves together with a correction factor dependent on the molar density were used in the CC-MW estimation software v1.3 [27,28]. APCI-DIP (atmospheric pressure chemical ionization-direct inlet probe) mass determinations were performed on a Finnigan LCQ Deca (*ThermoQuest*). ESI (electrospray ionization) mass spectra were recorded on a microTOF (Bruker Daltonics, Bremen, Germany). Mass calibration was carried out immediately before sample measurement on sodium formate clusters or by the ESI-Tune Mix standard (Agilent, Waldbronn, Germany). Elemental analyses were performed with a HEKAtech Euro EA CHNS elemental analyzer (Wegberg, Germany). Samples were prepared in a Sn cup and analyzed with added V_2_O_5_.

### 3.2. X-ray Diffraction Measurements

Crystallographic measurements were carried out on a *Stoe* IPDS2 or a *Stoe* StadiVari diffractometer with a STOE image plate detector and a Mo-Kα (λ = 0.71073 Å) monochromator or a *Stoe* StadiVari diffractometer with a Pilatus 200K image plate detector and Cu-Kα (λ = 1.54186 Å) radiation. Direct methods were used to solve the measurements and refined by “least-square” cycles (SHELXL-2014) [29]. All nonhydrogen atoms were anisotropically refined without restriction. Evaluation of the data sets and graphical preparation of the structures were carried out using Olex2 [30] and Mercury [31]. Details of the structure determination and refinement are summarized in Appendix A (supporting information).

### 3.3. Synthetic Protocols and Characterization

Synthesis of **3a**

To a solution of **1** (36 mg, 0.1 mmol) in 0.6 mL C_6_D_6_, 24 mg (0.1 mmol) of diphenyldisulfide was added at room temperature. This mixture was stirred at room temperature for 2 h. All volatile components were removed in vacuo, and the residue was subjected to flash column separation on silica using a mixture of pentane and methylene chloride (gradient) as eluent, which afforded 85% yield (49 mg, 0.09 mmol) as orange crystals.

Alternative: To a mixture of **2** (374 mg, 1 mmol) in 20 mL toluene, 437 mg (2 mmol) of diphenyldisulfide was added as a solution in 10 mL toluene at 0 °C. Under vigorous stirring, a few drops of tetrahydrofuran (THF) was added until all solid was dissolved. All volatile components were removed in vacuo, and the residue was subjected to flash column separation on silica using a mixture of pentane and methylene chloride (9/1) (R_f_ = 0.17) as eluent, which afforded 79% yield (458 mg, 0.79 mmol) as orange crystals. The following NMR chemical shifts, marked with an asterisk, indicate two distinct signals due to *rac*/*meso* diastereomers.

^1^H-NMR (400 MHz, C_6_D_6_): δ 7.87 (m, 4H, Ph), 7.09 (m, 4H, Ph), 6.95 (m, 2H, Ph), 4.52 (m, 1H, Cp), 4.42 (m, 1H, Cp), 4.31 (m, 1H, Cp), 4.27 (m, 1H, Cp), 4.21 (m, 3H, Cp), 4.19 (m, 1H, Cp), 1.05* and 1.05* (d, ^3^*J*_PH_ = 12.9 Hz and 12.8 Hz, 18H, *t*Bu CH_3_). ^13^C-NMR (101 MHz, C_6_D_6_): δ 137.3 (m, Ph C_q_), 131.4 (m, Ph), 129.2 (m, Ph), 126.3 (m, Ph), 77.8–77.3 (m, Cp C_ipso_), 77.0 (m, Cp), 76.5 (m, Cp), 76.5 (m, Cp), 72.7–72.2 (m, Cp), 72.1 (m, Cp), 33.4 (m, *t*Bu C_q_), 27.3 (m, *t*Bu CH_3_). ^31^P-NMR (202 MHz, C_6_D_6_): δ 51.2* (s), 50.2* (s). Elemental analysis (%): calculated: C 62.28, H 6.27, S 11.08, found: C 62.56, H 6.46, S 11.12. MS (APCI-DIP-HR) *m*/*z*: 579.115627 ([M + H]^+^ 100%), calculated: 579.115584.

Synthesis of **3b**

To a solution of **1** (64 mg, 0.18 mmol) in 1 mL toluene, 64 mg (0.2 mmol) of diphenyldiselenide was added at room temperature. This mixture was stirred at room temperature for 30 min. All volatile components were removed in vacuo, and the product was crystallized from a concentrated pentane solution at −20 °C, which afforded 95% yield (113 mg, 0.17 mmol) as orange crystals.

Alternative: To a mixture of **2** (374 mg, 1 mmol) in 20 mL toluene, 624 mg (2 mmol) of diphenyldisulfide was added as a solution in 10 mL toluene at 0 °C. Under vigorous stirring, a few drops of THF was added until all solid was dissolved. All volatile components were removed in vacuo, and the residue was subjected to flash column separation on silica using a mixture of pentane and methylene chloride (9/1) (R_f_ = 0.18) as eluent, which afforded 82% yield (548 mg, 0.82 mmol) as orange crystals. The following NMR chemical shifts, marked with an asterisk, indicate two distinct signals due to *rac*/*meso* diastereomers.

^1^H-NMR (400 MHz, C_6_D_6_): δ 7.94 (m, 4H, Ph), 7.05 (m, 4H, Ph), 6.98 (m, 2H, Ph), 4.42 (m, 1H, Cp), 4.36 (m, 1H, Cp), 4.31 (m, 1H, Cp), 4.28 (m, 2H, Cp), 4.24 (m, 3H, Cp), 1.08* and 1.07* (d, ^3^*J*_PH_ = 12.9 Hz and 12.8 Hz, 18H, *t*Bu CH_3_). ^13^C-NMR (101 MHz, C_6_D_6_): δ 137.3 (m, Ph C_q_), 131.4 (m, Ph), 129.2 (m, Ph), 126.3 (m, Ph), 77.4 (m, Cp), 76.9 (d, ^1^*J*_CP_ = 32 Hz, Cp C_ipso_), 76.9 (m, Cp), 76.7 (m, ^1^*J*_CP_ = 33 Hz, Cp C_ipso_), 73.8–73.6 (m, Cp), 73.0–72.7 (m, Cp), 33.0 (m, *t*Bu C_q_), 27.8 (m, *t*Bu CH_3_). ^31^P-NMR (202 MHz, C_6_D_6_): δ 57.8* (s), 57.1* (s). ^77^Se-NMR (95 MHz, C_6_D_6_): 216.4* (d, ^1^*J*_PSe_ = 225 Hz, ^n^*J*_PSe_ = 5 Hz), 215.2* (d, ^1^*J*_PSe_ = 224 Hz). Elemental analysis (%): calculated: C 53.59, H 5.40, found: C 53.61, H 5.42. MS (APCI-DIP-HR) *m*/*z*: 675.005211 ([M + H]^+^ 100%), calculated: 675.004486.

Synthesis of **3c**

To a solution of **1** (360 mg, 1 mmol) in 10 mL of THF diphenylditelluride (410 mg, 1 mmol) was added at room temperature, and the mixture was stirred at 90 °C overnight in a pressure Schlenk tube. All volatile components were removed in vacuo, and the product was crystallized from a concentrated pentane solution at −20 °C, which afforded the crude product as orange crystals. The product contained a significant amount of starting material (diphenylditelluride and diphospha-[2]ferrocenophane **1**). The following NMR chemical shifts, marked with an asterisk, indicate two distinct signals due to *rac*/*meso* diastereomers.

^1^H-NMR (400 MHz, C_6_D_6_): δ 8.04 (m, 4H, Ph), 6.99 (m, 6H, Ph), 4.44 (m, 1H, Cp) 4.37 (m, 1H, Cp), 4.34 (m, 1H, Cp), 4.30 (m, 1H, Cp), 4.25 (m, 3H, Cp), 4.00 (m, 1H, Cp), 1.12* and 1.11* (d, ^3^*J*_PH_ = 12.6 Hz and 12.3 Hz, 18H, *t*Bu CH_3_). ^13^C-NMR (101 MHz, C_6_D_6_): δ 138.3 (m, Ph), 129.6 (m, Ph), 127.5 (m, Ph), 112.1 (m, Ph C_q_), 78.0 (m, Cp), 77.5 (m, Cp), 77.1 (m, Cp), 76.9 (m, Cp), 76.2 (m, ^1^*J*_CP_ = 25 Hz, Cp C_ipso_), 75.9 (m, ^1^*J*_CP_ = 26 Hz, Cp C_ipso_), 73.5 (m, Cp), 32.3 (m, *t*Bu C_q_), 28.8 (m, *t*Bu CH_3_). ^31^P-NMR (202 MHz, C_6_D_6_): δ 45.8* (s), 45.3* (s). ^125^Te-NMR (158 MHz, C_6_D_6_): δ 288.2* (dd, ^1^*J*_PTe_ = 497 Hz, ^n^*J*_PTe_ = 14 Hz), 284.2* (d, ^1^*J*_PTe_ = 497 Hz). MS (APCI-DIP-HR) *m*/*z*: 806.974229 ([M + H + 2O]^+^ 17%), calculated: 806.973718.

Synthesis of **4a**

To a solution of **1** (32 mg, 0.09 mmol) in 0.6 mL C_6_D_6_, 0.1 mL (1.1 mmol) of dimethyldisulfide was added at room temperature. This mixture was stirred overnight at 90 °C. All volatile components were removed in vacuo, and the residue was subjected to flash column separation on silica using a mixture of pentane and methylene chloride as eluent, which afforded 87% yield (35 mg, 0.08 mmol) as orange crystals. The following NMR chemical shifts, marked with an asterisk, indicate two distinct signals due to *rac*/*meso* diastereomers.

^1^H-NMR (400 MHz, C_6_D_6_): δ 4.46 (m, 2H, Cp), 4.38 (m, 6H, Cp), 2.23 (m, ^3^*J*_PH_ = 13.1 Hz 6H, S-CH_3_), 1.07 (d, ^3^*J*_PH_ = 12.7 Hz 18H, *t*Bu CH_3_). ^13^C-NMR (101 MHz, C_6_D_6_): δ 78.2–77.6 (m, Cp C_ipso_), 76.5 (m, Cp), 76.1 (m, Cp), 72.5–72.2 (m, Cp), 33.2 (m, *t*Bu C_q_), 27.1 (m, *t*Bu CH_3_), 17.8 (m, ^2^*J*_PC_ = 32 Hz, S-CH_3_), 17.7 (d, ^2^*J*_PC_ = 32 Hz, S-CH_3_). ^31^P-NMR (202 MHz, C_6_D_6_): δ 56.8* (s), 56.5* (s). Elemental analysis (%): calculated: C 52.87, H 7.10, S 14.11, found: C 53.75, H 7.57, S 13.81. MS (APCI-DIP-HR) *m*/*z*: 455.084305 ([M + H]^+^ 100%), calculated: 455.084284.

Synthesis of **4b**

To a solution of **1** (72 mg, 0.2 mmol) in 2 mL toluene, 0.1 mL (1.0 mmol) of dimethyldiselenide were added at room temperature. This mixture was stirred at room temperature for 3 h. All volatile components were removed in vacuo, and the residue was subjected to flash column separation on silica using a mixture of pentane and methylene chloride as eluent affording 73% yield (80 mg, 0.15 mmol) as an orange oil, which transformed over time to a yellow solid. The following NMR chemical shifts, marked with an asterisk, indicate two distinct signals due to *rac*/*meso* diastereomers.

^1^H-NMR (400 MHz, C_6_D_6_): δ 4.44–4.37 (m, 7H, Cp), 4.33 (m, 1H, Cp), 2.09 (m, ^3^*J*_PH_ = 10.0 Hz 6H, Se-CH_3_), 1.10 (d, ^3^*J*_PH_ = 12.5 Hz 18H, *t*Bu CH_3_). ^13^C-NMR (101 MHz, C_6_D_6_): 77.1 (m, ^1^*J*_PC_ = 30 Hz, Cp C_ipso_), 77.0 (m, Cp), 76.5 (m, Cp), 73.9–73.7 (m, Cp), 72.9 (m, Cp), 72.8–72.6 (m, Cp), 32.6 (m, *t*Bu C_q_), 27.5 (m, *t*Bu CH_3_), 6.5 (d, ^2^*J*_PC_ = 30 Hz, Se-CH_3_), 6.4 (d, ^2^*J*_PC_ = 30 Hz, Se-CH_3_). ^31^P-NMR (202 MHz, C_6_D_6_): δ 58.7* (s), 58.4* (s). ^77^Se-NMR (95 MHz, C_6_D_6_): −21.5* (d, ^1^*J*_PSe_ = 218 Hz), −21.8* (dd, ^1^*J*_PSe_ = 218 Hz, ^n^*J*_PSe_ = 6 Hz). Elemental analysis (%): calculated: C 43.82, H 5.88, found: C 43.68, H 6.20. MS (APCI-DIP-HR) *m*/*z*: 550.973667 ([M + H]^+^ 100%), calculated: 550.973186.

Synthesis of **5**

To a solution of **3b** (67 mg, 0.1 mmol) in 1 mL DCM, 31 mg (0.1 mmol) of [Cu(MeCN)_4_][BF_4_] was added at room temperature. This mixture was placed in the ultrasonic bath for 1 h. All volatile components were removed in vacuo, and the residue was dissolved in 10 mL toluene. The solid was removed by centrifugation, and the product was crystallized from a concentrated toluene solution at −20 °C, which afforded 35% yield (30 mg, 0.03 mmol) as orange crystals.

^1^H-NMR (400 MHz, CD_3_CN): δ 7.87 (m, 4H, Ph), 7.43 (m, 6H, Ph),4.76 (m, 2H, Cp), 4.54 (m, 2H, Cp), 4.51 (m, 2H, Cp), 4.39 (m, 2H, Cp), 1.96 (s, 3H, MeCN), 0.95 (m, 18H, *t*Bu CH_3_). ^13^C-NMR (101 MHz, CD_3_CN): δ 137.1 (m, Ph), 131.3 (s, Ph), 130.4 (m, Ph), 127.1 (m, Ph C_q_), 118.3 (s, Me-CN), 78.1 (m, Cp), 75.3 (m, Cp), 75.2 (s, Cp), 74.0 (m, Cp C_ipso_), 72.8 (s, Cp), 35.4 (m, *t*Bu C_q_), 26.8 (m, *t*Bu CH_3_), 1.7 (s, H_3_C-CN). ^31^P-NMR (202 MHz, CD_3_CN): δ 51.2 (s). ^77^Se-NMR (95 MHz, CD_3_CN): δ 233.4–228.1 (m). Elemental analysis (%): calculated: C 44.50, H 4.55, N 1.62 found: C 43.58, H 4.64, N 1.72. MS (ESI-HR) *m*/*z*: 736.926803 ([M − (BF_4_) − (MeCN)]^+^ 100%), calculated: 736.926262.

Synthesis of **6**

To a solution of **3b** (67 mg, 0.1 mmol) in 1 mL MeCN, 20 mg (0.1 mmol) of AgBF_4_ was added at room temperature. This mixture was placed in the ultrasonic bath for 1 h. All volatile components were removed in vacuo, and the residue was washed with 10 mL toluene. The product was crystallized from a concentrated MeCN solution at −20 °C, which afforded quantitative yield (91mg, 0.1 mmol) as yellow crystals.

^1^H-NMR (400 MHz, CD_3_CN): δ 7.84 (m, 4H, Ph), 7.44 (m, 6H, Ph), 4.76 (m, 2H, Cp), 4.53 (m, 2H, Cp), 4.51 (m, 2H, Cp), 3.97 (m, 2H, Cp), 1.94 (m, MeCN), 0.93 (m, 18H, *t*Bu CH_3_). ^13^C-NMR (101 MHz, CD_3_CN): δ 136.3 (m, Ph), 131.6 (m, Ph), 130.3 (m, Ph), 130.1 (m, Ph C_q_), 118.4 (s, Me-CN), 78.7 (m, Cp), 75.0 (m, Cp), 75.0 (m, Cp), 72.8 (s, Cp), 35.4 (m, *t*Bu C_q_), 27.0 (m, *t*Bu CH_3_), 1.5 (m, H_3_C-CN). ^31^P-NMR (202 MHz, CD_3_CN): δ 75.6 (m, ^1^*J*_P-(109)Ag_ = 462 Hz, ^1^*J*_P-(107)Ag_ = 401 Hz). ^77^Se-NMR (95 MHz, CD_3_CN): δ 230.1–225.1 (m, ^1^*J*_PSe_ = 286 Hz). Elemental analysis (%): calculated: C 42.33, H 4.33, N 1.54 found: C 42.16, H 4.33, N 1.37. MS (ESI-HR) *m*/*z*: 780.902890 ([M − (BF_4_) − (MeCN)]^+^, calculated: 780.901754.

Synthesis of **7**

To a solution of **3b** (67 mg, 0.1 mmol) in 2 mL toluene, 64 mg (0.2 mmol) of (tht)AuCl was added at room temperature. The mixture was stirred for 1 h at room temperature. All volatile components were removed in vacuo, and the residue was washed with 5 mL toluene. The product was crystallized from a concentrated toluene/THF solution at −20 °C, which afforded 92% yield (105 mg, 0.09 mmol) as yellow crystals. The following NMR chemical shifts, marked with an asterisk, indicate two distinct signals due to *rac*/*meso* diastereomers.

^1^H-NMR (400 MHz, CD_2_Cl_2_): δ 8.01 (m, 2H, Ph), 7.50 (m, 2H, Ph), 7.45 (m, 4H, Ph), 4.81–4.75 (m, 6H, Cp), 4.63 (m, 1H, Cp), 4.60 (m, 1H, Cp), 1.18* and 1.17* (d, ^3^*J*_PH_ = 19.3 Hz and 19.2 Hz, 18H, *t*Bu CH_3_). ^13^C-NMR (101 MHz, CD_2_Cl_2_): δ 137.6 (m, Ph), 130.6 (m, Ph), 130.3 (m, Ph), 127.2 (m, Ph C_q_), 77.5 (m, Cp), 76.0 (m, Cp), 75.6 (m, Cp), 74.4 (m, Cp), 72.7 (m, ^1^*J*_CP_ = 23 Hz, Cp C_ipso_), 38.3 (m, *t*Bu C_q_), 26.9 (m, *t*Bu CH_3_). ^31^P-NMR (202 MHz, CD_2_Cl_2_): δ 85.9* (s), 85.2* (s). ^77^Se-NMR (95 MHz, CD_2_Cl_2_): δ 271.1* (d, ^1^*J*_PSe_ = 330 Hz), 268.9* (d, ^1^*J*_PSe_ = 332 Hz). Elemental analysis (%): calculated: C 31.69, H 3.19 found: C 31.73, H 3.21. MS (ESI-HR) *m*/*z*: 1102.898926 ([M − Cl]^+^ 100%), calculated: 1102.898618.

Synthesis of **8**

For this, 67 mg of **3b** (0.1 mmol) and 32 mg (0.1 mol) of (tht)AuCl were dissolved in 0.6 mL σ-difluorobenzene. The solution was added to 14 mg (0.1 mmol) AlCl_3_ at room temperature, and the resulting suspension was placed into the ultrasonic bath until all solid was dissolved. It was crystallized overnight at room temperature. The crystals were separated from the solution and washed with 3 mL of toluene, which afforded 46% yield (96 mg, 0.05 mmol) as yellow crystals.

^1^H-NMR (400 MHz, CD_2_Cl_2_): δ 7.85 (m, 8H, Ph), 7.55 (m, 4H, Ph), 7.46 (m, 8H, Ph), 5.25 (m, 4H, Cp), 4.84 (m, 4H, Cp), 4.81 (m, 4H, Cp), 4.77 (m, 4H, Cp), 1.03 (m, 36H, *t*Bu CH_3_). ^13^C-NMR (101 MHz, CD_2_Cl_2_): δ 136.9 (s, Ph), 131.3 (s, Ph), 131.0 (s, Ph), 124.4 (s, Ph C_q_), 77.7 (m, Cp), 75.6 (m, Cp), 74.8 (m, Cp), 74.0 (m, Cp), 72.3 (m, Cp C_ipso_), 38.8 (m, *t*Bu C_q_), 26.9 (m, *t*Bu CH_3_). ^27^Al-NMR (104 MHz, CD_2_Cl_2_): δ 103.7 (s). ^31^P-NMR (202 MHz, CD_2_Cl_2_): δ 91.0 (s). ^77^Se-NMR (95 MHz, CD_2_Cl_2_): δ 275.3 (m, ^1^*J*_PSe_ = 177 Hz). Elemental analysis (%): calculated: C 34.71, H 3.50 found: C 34.71, H 3.53. MS (ESI-HR) *m*/*z*: 870.963213 ([1/2(M − AlCl_4_)]^+^ 76%), calculated: 870.963937.

## 4. Conclusions

In summary, we report on a series of unprecedented ferrocene-bridged bischalcogenophosphinous acid esters with the chalcogens S, Se, and Te. For tellurophosphinous acid, this is the first example of a molecular structure established via single X-ray diffraction. The compounds are generally obtained as mixtures of *rac* and *meso* diastereomer where the ratio is not affected by the stereochemical orientation of the precursor, for instance, starting from chiral [2]ferrocenophane **1** or prochiral bisphosphanide **2**. Investigation of the coordination behavior of bisselenophosphinous acid ester toward d^10^ coinage metal centers revealed exclusive metal coordination at the lone pair at phosphorus based on X-ray crystallography. Interestingly, the coordination shift of the ^77^Se NMR resonance was equally or more sensitive to the nature of the metal than the corresponding ^31^P NMR resonance although the latter atom was directly attached to the metal center. For ^77^Se, coordination of M(I) (M = Cu, Ag, and Au) deshielding was observed in all cases (Cu: +14 ppm, Ag: +11 ppm, and Au: +55 ppm). For ^31^P, shielding was observed for Cu (−6 ppm), as in related cases [26,32,33], but deshielding was observed for Ag and Au coordination (Ag: +18 ppm and Au: +28 ppm). An intriguing feature of *meso* bischalcogenophosphinous acid esters with Se and Te was the observation of a resolved coupling of ^77^Se and ^125^Te to both chemically equivalent but magnetically inequivalent phosphorus nuclei within the molecule. This finding is interpreted as a consequence of attractive interactions of the chalcogenophenyl or chalcogenomethyl units in solution, most likely directly via the chalcogen atoms (Ch···Ch).

## Data Availability

The data presented in this study are available in the Appendix A.

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
