# Peer review of "C2-Symmetric P-Stereogenic Ferrocene Ligands with Heavier Chalcogenophosphinous Acid Ester Donor Sites†"

_molecules, 2021, doi:10.3390/molecules26071899_

Round 1

Reviewer 1 Report

In this manuscript the authors report the efficient synthesis of ferrocene bridged bischalcogenophosphinous acid esters through reaction of tert-butyl-substituted diphospha[2]ferrocenophane with dichalcoganes. The coordination behaviour of the seleno derivative towards coinage metal centers has been also investigated. Mononuclear complexes were obtained in the case of Cu(I) and Ag(I). However, because of the preferred linear coordination of Au(I), the reaction with (tht)AuCl led to a dinuclear complex. In all the cases exclusive metal coordination at phosphorous was found. The resulting complexes have been characterized by NMR and X-ray diffraction. Notably, the authors report the first solid-state structure of a telluro phosphinous acid ester.

In my opinion, this manuscript represents an interesting contribution in the synthesis of new ferrocene derivatives with potential applications. Some of the complexes prepared could even display interesting catalytic properties. In summary, I support publication of this manuscript in Molecules.

A minor issue: In the characterization of the reported products, the signal corresponding to the tert-butyl group has been described in some cases as multiplet. However, I think that it is better described as doublet (in fact the authors report the coupling constant).

Author Response

cf. attached file

Reviewer 2 Report

The contribution by Pietschnig et al. is interesting and scientifically sound. I recommend acceptance with minor revision along the following lines:

  • Title: P-stereogenic would be better than P-chiral. Chirality is not a property of an atom.
  • I understand that the compounds prepared were obtained as diastereomeric meso/rac However, the NMR spectra show only one set of signals. Which diastereomer is reported and what are the analytical data of the other one?
  • Line 66: mixtures
  • Line 68: ... have been known for several decaes.
  • Scheme 3, iii: Why did the authors use 2 equiv. of gold salt? What is the result with only one and no Lewis acid?
  • Line 201: (Scheme 4)
  • Experimental part: No IR spectra? What about cyclovoltammograms, in particular of the bi- and tetrametallic complexes?
  • Line 266: pentane and methylene chloride (X/Y) - give ratio
  • References: Abbrevitions according to CASSI should be used.
  • 23 (Line 485): Is the journal Chem Eur. J.?

Author Response

cf. attached file
